# 2CA-R^2^: A Hybrid MAC Protocol for Machine-Type Communications

**DOI:** 10.3390/s25102994

**Published:** 2025-05-09

**Authors:** Sergio Javier-Alvarez, Pablo Hernandez-Duran, Miguel Lopez-Guerrero, Luis Orozco-Barbosa

**Affiliations:** 1Graduate Program on Information Science and Technology, Metropolitan Autonomous University (Iztapalapa), Mexico City 09340, Mexico; serjaval@xanum.uam.mx (S.J.-A.); dhernandez@cultura.gob.mx (P.H.-D.); 2Department of Electrical Engineering, Metropolitan Autonomous University (Iztapalapa), Mexico City 09340, Mexico; milo@xanum.uam.mx; 3Albacete Research Institute of Informatics, Universidad de Castilla-La Mancha, 02006 Albacete, Spain

**Keywords:** machine-to-machine communications, medium access control, wireless LAN, protocols, standards

## Abstract

Machine-to-machine (M2M) communications are becoming the most important factor shaping network traffic. However, traditional controls developed for human-generated traffic are not able to cope with new demands. Thus, hybrid MAC protocols have been proposed to make use of the combined advantages of contention and reservation. Most of them are based on a contention stage (where a variant of CSMA/CA or ALOHA is used) followed by a reservation stage (e.g., TDMA or FDMA). In this paper, we introduce 2CA-R^2^, a hybrid MAC protocol for M2M communications intended to be used in the device domain. What distinguishes this proposal is that the contention stage is controlled by a conflict–resolution algorithm known as Adaptive-2C. The protocol was evaluated using a model based on a Markov chain and computer simulations. Its performance was compared with DCF, the MAC technique used in IEEE802.11 standards. Our results show significant improvements over DCF in various metrics of network performance across different traffic situations. We also evaluated the time the protocol takes to resolve an access conflict, and we observed substantial improvements in the number of stations that can be served with the same network resource (in some cases, around a 40% improvement).

## 1. Introduction

Some recent studies [1] predict that by 2050, most of the population in the world will live in urban environments; therefore, cities will have to efficiently manage their resources and provide services more effectively. Due to such transition toward smart cities, the Internet is evolving to become a ubiquitous network of interconnected devices that will allow the monitoring and control of multiple environmental variables in real time. This trend, where the vast majority of Internet users are machines and not humans, signals the arrival of the Internet of Things (IoT). Such an exponential growth will also bring about the need to provide wireless network access to a large number of autonomous devices. In fact, a recent study conducted by IoT Analytics [2] predicts that by 2030, there will be 41.1 billion active IoT device gateways, or access points, worldwide. This situation has motivated the evolution of diverse technologies, mainly cellular communications and wireless local area networks, with the consideration that a large number of connected devices may need network access services at one time.

For this vision of the future to be achieved, adequate support for M2M communications is indispensable. However, the available studies provide evidence to support the fact that the characteristics of traffic generated in access networks by IoT applications are different from those present in traditional Human-Type Communications (HTC) [3,4]. For example, in M2M communications, data are predominantly transferred in the uplink direction, whereas in HTC it is otherwise. Other characteristics that can be identified in M2M communications are the following: First, in many IoT applications, the effective payload of data packets is small [5]. Second, these packets can be generated not only at random instants but also at regular time intervals. A networking technology that intends to serve M2M communication devices must implement strategies that take into account the characteristics of this type of traffic. However, existing protocols, which were developed for HTC, are inefficient under such conditions [6].

MAC protocols are critical components of a network architecture intended to be used in IoT applications. In a typical scenario, these protocols must be designed considering that a large number of IoT devices may simultaneously attempt to transmit small packets over low-data-rate links. Therefore, MAC algorithms must be scalable, that is, when the number of simultaneous access requests grows, a number of desirable performance features should be preserved (e.g., high throughput, low latency, and fairness). However, widely used MAC protocols, which were conceived to support human-to-human or human-to-machine interactions, have limitations in this regard. In addition, IoT devices are typically both powered by a battery and limited in terms of processing power; therefore, such protocols should also be simpler to implement and have lower computational costs compared to protocols used in other applications. For all these reasons, in recent years, we have seen the appearance of several MAC proposals intended to be used in M2M communications to improve performance or incorporate additional features.

At this point, it is worth revisiting some basics. MAC protocols can be broadly classified as contention-based, reservation-based, and hybrid. Contention-based MAC protocols are the simplest in terms of configuration and implementation. Among these protocols, we have the following well-known examples [7,8]: ALOHA, slotted ALOHA (s-ALOHA), and Carrier Sense Multiple Access with Collision Avoidance (CSMA/CA). An implementation of the CSMA/CA protocol is the basis of the Distributed Coordination Function (DCF), which is the fundamental access method of the popular IEEE 802.11 standard [9] for wireless local area networks (WLANs).

A disadvantage of contention-based MAC protocols is that as the number of devices in the network grows, the collision probability also increases. Although some control measures can be taken to mitigate this situation, this factor leads to limited scalability [10,11]. Other effects that arise under such circumstances are an increased waste of power coming from both collisions and idle listening and greater overhead due to the transmission of control packets, which can consume more power than the actual transmission of data packets [10]. All of these effects are further exacerbated when simultaneous access attempts occur in large numbers.

Reservation protocols, on the other hand, eliminate the problems caused by collisions by preallocating transmission resources to network stations. Among these solutions, we can find the following two well-known cases [7,8]: Time Division Multiple Access (TDMA) and Frequency Division Multiple Access (FDMA).

A typical problem associated with reservation protocols is that they exhibit limited adaptability to changing conditions in network traffic. In TDMA, for example, if a device runs out of data to transmit, its reserved transmission time slots will not be used, which leads to a waste of resources. Another issue to be taken into account is the overhead generated by the resource allocation procedure. In addition, reservation protocols have a number of technical challenges that must be addressed. For example, TDMA requires strict clock synchronization to avoid communication failures due to interference among devices [11]. This feature causes additional power consumption since it requires the regular transmission and detection of a clock signal [12]. In turn, FDMA is poorly suited for operation with low-cost IoT devices, as they require complex circuitry to communicate and switch between different radio channels.

Hybrid MAC protocols belong to a third category, which is intended to incorporate the best features of both contention and reservation protocols, providing more efficient medium access control. One possible approach to implement a hybrid protocol is based on the division of each transmission cycle into at least two stages, or periods, namely, contention and data transmission. In the former, the nodes compete for the opportunity to transmit, typically using a contention protocol. The nodes that are successful in this period will receive the resource allocation (e.g., a time slot or a channel frequency) to transmit their data during the latter.

Generally speaking, hybrid protocols provide greater flexibility than pure reservation protocols, and they scale better than pure contention protocols. This is the key motivation for developing this approach for channel sharing, since both characteristics are important for M2M communications. Therefore, it is not surprising that in recent years we have witnessed the appearance of a number of hybrid protocols intended to be used in M2M communications (e.g., see [11,12]).

In spite of their advantages, hybrid MAC protocols are not exempt from technical challenges, since they also inherit the limitations of their constituent protocols. A problem that arises in some contention-based MAC protocols, regardless of whether they are used in isolation or as part of a hybrid protocol, is related to the way they try to maintain a small collision probability. Some popular protocols, such as DCF, resolve contention by means of random access controlled by the well-known binary exponential backoff algorithm. In this approach, the stations draw a random turn from a set (i.e., the “contention window”) whose size geometrically grows with each collision [13]. Therefore, when there is a large number of contending stations, this strategy can lead to long waiting times and a waste of resources. In addition, the fact that the contention window is locally updated (i.e., at each node) may also lead to a certain degree of unfairness in the way access is granted to the network nodes. Under heavy traffic conditions, a node may experience a number of collisions and, in consequence, large delays (due to the growth of its contention window) before its packet goes through. At the same time, another node that all of a sudden joins the contention (with a small contention window) may successfully transmit its packet on its first try. This effect is not desirable in many applications, especially the ones that require high-priority data delivery. Therefore, optimization of the contention algorithm used in hybrid protocols is a relevant matter that deserves attention. An alternative, which provides low access delay when collisions occur in the medium, is the implementation of protocols related to the tree algorithm [6,14]. An example of this approach is Distributed Queuing (DQ), which was introduced in [15], and a variant was presented in [6]. Another variant of the tree algorithm, known as the two-cell algorithm (2C) [16], shows some level of fairness with the following rule. When a collision occurs, no new nodes are allowed to join the contention until all the ones involved in the previous collision successfully transmit.

In this paper, we introduce 2C (adaptive) with resource reservation, or 2CA-R^2^ for short, a hybrid MAC protocol that combines random access to achieve resource reservation and transmission based on time scheduling. The proposal consists of two stages, namely, “collision resolution interval” and “data transmission interval”. The proposed 2CA-R^2^ protocol is intended to support M2M communications in the device domain [17], considering a network within the transmission range of a WLAN or shorter. What distinguishes the proposal from other works is the use of the Adaptive-2C conflict resolution algorithm as the random access mechanism in the contention stage. The Adaptive-2C algorithm was previously introduced and evaluated by our research team in [18]. As it will be described in the following, the properties of this contention algorithm allow us to achieve a seamless integration between the two intervals, i.e., contention and transmission. When the collision resolution interval ends, the stations take turns to transmit their data packets using contention-free slots.

The main contribution of this paper can be summarized as the introduction and evaluation of 2CA-R^2^, a hybrid adaptive 2C-based MAC protocol. The evaluation was carried out using both computer simulations and a mathematical model based on a Markov chain. Although this protocol is simple to implement, our results demonstrate that it scales better and achieves outstanding performance when compared to the widely used DCF protocol.

The remainder of this paper is structured as follows. In Section 2, we provide an overview of some related works. In Section 3, we describe the base protocols, namely, 2C and Adaptive-2C. In Section 4, we describe the proposed hybrid protocol, that is, 2CA-R^2^. In Section 5, we derive an analytical model intended to estimate the performance of the Adaptive-2C algorithm, which is the basis of the collision resolution interval of the proposal. In Section 6, we describe the performance evaluation experiments that were carried out and discuss the corresponding results. Finally, Section 7 concludes this paper.

## 2. Related Work

In the literature, there are several pieces of work whose purpose is to introduce hybrid MAC mechanisms for M2M communications. This section provides an overview of some relevant works on this subject, but we focus on protocols whose operation is similar to our proposal, that is, protocols that are divided into at least the two previously mentioned stages: contention and transmission. Note that the elements of a description should not be used to describe other works since there is no guarantee that they mean the same concept.

In [19], Liu et al. propose a hybrid MAC protocol for M2M networks. In the proposal, there is a contention stage, called contention only period (COP), and a data transmission stage, called transmission only period (TOP). During the COP, which is based on *p*-persistent CSMA, the devices compete for transmission slots. Before the TOP takes place, the IDs of successfully contending devices and scheduling information are transmitted by a base station in an announcement period (AP). Only successful devices can transmit data during the TOP using TDMA. Analytical and simulation results shown in [19] demonstrate the advantages of this hybrid MAC protocol when compared to ALOHA, s-ALOHA, and TDMA. However, significant control and computational overhead may arise, as the base station must transmit synchronization frames, the IDs of successful devices, and scheduling information during the AP.

The framework implemented in the hybrid MAC protocol proposed by Verma et al. in [20] combines CSMA, DCF, and TDMA. The proposal consists of four different stages: a notification period (NP), a contention period (CP), an announcement period (AP), and a transmission period (TP). First, the access point broadcasts a notification to all devices during NP to start the CP. During the CP, the devices that have data to transmit send transmission requests to the access point using the CSMA technique. Then, during the AP, the access point signals the end of the contention period and the beginning of the TP to the nodes whose transmission requests where successful. Finally, during the TP, the devices transmit data using the DCF mode of IEEE 802.11 within TDMA slots to cope with communication failures due to the lack of clock synchronization. In spite of the evident signaling overhead, the authors report that the proposed MAC protocol achieves better performance than slotted-ALOHA and TDMA in terms of aggregate throughput and average transmission delay.

In the hybrid MAC protocol proposed for M2M networks by Hegazy et al. [21], each frame consists of three main parts: contention only period (COP), notification only period (NOP), and transmission only period (TOP). During the COP, the devices contend for transmission time slots using the non-persistent CSMA (NP-CSMA) mechanism. In the NOP, the base station announces the time slot reservations to be used in the TOP to transmit data packets. The method used for transmission is TDMA. It is interesting to note that if a successfully contending device cannot be granted a time slot during the TOP, it can send its data within the transmission slot of another device subject to the restriction that a target Signal-to-Interference and Noise Ratio (SINR) is not exceeded. In this way, the mechanism increases the number of transmitted packets. The authors report that the proposed protocol achieves higher throughput, lower average delay time, and a lower packet collision ratio when compared with NP-CSMA and s-ALOHA.

Yang et al. [22] proposed a scalable MAC framework assisted by machine learning. In the work, each cycle, called a superframe, consists of four parts: a rendezvous period (RP), a contention access period (CAP), a notification period (NP), and a contention-free period (CFP). A particular characteristic of this proposal is that by using machine learning techniques, in the RP a gateway dynamically detects the number of active devices from the signals that they emit. According to the number of active devices that have been detected, the gateway can determine the optimal length (*M*, measured in time slots) of the CAP. Then, each active device randomly selects one of the *M* slots and sends a request message to contend for a data slot to be used in the CFP. If two or more devices select the same time slot in the CAP, a collision occurs. In such a case, they have to wait for a new frame to retry transmission. Once the request messages have been received, the gateway allocates data slots to the successful devices. This information is announced in the NP. During the CFP, the devices that have been allocated data slots can transmit data packets using the TDMA policy. In spite of the increased complexity of this framework, extensive simulations reported in [23] demonstrated that it achieves superior performance when compared to solutions like s-ALOHA and reservation-based MAC protocols, such as TR-MAC [24]. However, since offline training is required, managing such a network becomes more challenging. In addition, although the value of *M* is optimally calculated in each cycle, the random selection of slots by the contending devices offers no guarantee of a low collision rate during the contention phase.

In the hybrid MAC protocol suggested by Saad et al. in [25] each frame consists of four periods: notification period (NP), contention period (CP), announcement period (AP), and transmission period (TP). At the beginning of each frame, in the NP, all devices receive a broadcast message from the base station (BS) which is intended to indicate the start of the CP. During the CP, devices that have a data packet to transmit contend by sending a transmission request (REQ) message to the BS to reserve a transmission slot using the s-ALOHA mechanism. If no collision occurs, the BS will reserve a valid time slot in the TP for the device whose REQ was successful. An ACK message is returned by the BS to notify this event. After the CP finishes, the AP starts. In it, the BS sends a message to all successful devices with the number of their reserved time slots. After that, these devices transmit their packets during the TP using TDMA. Saad et al. report that their proposal shows superior performance in terms of system throughput, average packet delay, success access ratio, and reservation ratio in comparison with some other hybrid schemes (i.e., NP-CSMA/TDMA, *p*-persistent CSMA/TDMA, and s-ALOHA/TDMA).

A hybrid MAC protocol suggested by Lachtar et al. in [26] classifies the devices into different classes depending on their priority level. It also divides time into cycles, where each one of them consists of three periods: a notification period (NP), a contention period (COP), and a transmission period (TOP). Each cycle is initiated with an NP, where a base station announces the start of the COP to all devices regardless of its class. Throughout the COP stage, the nodes with data to transmit use *p*-persistent CSMA to send transmission requests to the BS. Here, the devices belonging to a class with higher priority will have a higher probability of successful contention than devices belonging to a class with a lower priority. Successful nodes are granted a slot to transmit data using TDMA in the TOP. This architecture turns out to achieve higher throughput, lower average delay, and lower power consumption when compared with TDMA and *p*-persistent CSMA [26].

Other proposals were introduced by Olatinwo et al. in [27,28]. In [27], they introduce a hybrid multiclass MAC protocol for wireless body area networks (WBANs) whose framework consists of the following four stages: notification phase (NP), contention phase (CP), announcement phase (AP), and transmission phase (TP). At the beginning of a frame, in the NP, all devices in the network receive a message from the access point notifying them about the beginning of the CP. In the CP, only the devices that are ready to transmit contend for transmission opportunities by using the s-ALOHA scheme, and the successfully contending devices send their packets by employing a TDMA scheme in the TP. As a particularity of this protocol, the devices are grouped into two classes, where class 1 devices generate critical data that require high reliability and low delay, while class 2 devices generate packets that are less critical. Based on the class of each device and its data packet size, the coordinator (i.e., the access point) will take the necessary decisions to allocate resources during the TP. From the simulation results, it is concluded that the proposed protocol performs better than a protocol based on the combination of NP-CSMA and TDMA in terms of the system sum throughput and average packet delay. However, because s-ALOHA is used as the contention mechanism, the applications of this protocol seem to be more appropriate to long-range networks with low data rates. Otherwise, congestion scenarios may occur, leading to low scalability.

Furthermore, Olatinwo et al. proposed another approach in [28]. This one is based on a hybrid MAC frame for WBAN networks that consists of the following two stages: a contention period, called the “CSMA/CA period”, and a transmission period, called the “TDMA period”. At the beginning of the CSMA/CA period, the access point (AP) sends a message to all devices in the network. From that moment on, the devices that have data to transmit send a request-to-transmit (REQ-T) message at random but based on the size of their own contention window. If two or more devices send a REQ-T simultaneously, a collision will occur. Devices that do not have packets to transmit will enter into a sleep state to save power. To end the CSMA/CA period, the AP transmits a general feedback message (OACK) to all devices in order to inform which REQ-T messages were successful instead of sending individual acknowledgement messages each time it successfully receives a REQ-T message. Furthermore, the OACK message contains the transmission order so that a specific time slot for data transmission is assigned to each device with a successful REQ-T. During the TDMA period, data packet transmission is performed depending on the order established in the CSMA/CA period. In case of a successful transmission of a data packet, the AP returns an ACK message. Otherwise, this last message is not sent. In this way, the device whose packet was not delivered sends a retransmission intent message with which the AP will assign a new slot. The results reported in [28] show that the use of a single OACK leads to shorter delays when compared to the conventional ACK used in most of the literature. Furthermore, they concluded that their proposal achieves better performance than the HyMAC [29] and CPMAC [30] protocols. In spite of these promising results, and probably due to its intended application, the proposal was tested with a maximum of 15 devices attempting to transmit simultaneously. Thus, its scaling properties remain unknown.

In the hybrid MAC protocol proposed by Fan et al. in [31], each superframe is composed of five periods: beacon period (BP), contention period (CP), assignment period (AP), energy period (EP), and variable upload period (vUP). During the BP, the coordinator node broadcasts a synchronization packet containing essential parameters, such as the total number of minislots, a minislot duration and the maximum data packet length. In the CP, devices with data to transmit divide their payload into smaller packets (if necessary) and compete for channel access using a CSMA/CA-based algorithm. Each device is allowed to transmit a single packet that includes information about how many packets remain to be sent. Then, during the AP, the coordinator node broadcasts an assignment list with the reserved slots for each device that successfully transmitted during the CP. This assignment is based on the remaining number of packets indicated by each device. In the EP, the coordinator sends energy packets using radio frequency transfer technology to recharge the device batteries, which can harvest this energy while remaining in a sleep state. Finally, during the vUP, the devices transmit their remaining packets using the assigned slots. Fan et al. reported that their proposal, called PAH-MAC, shows significant advantages compared to protocols such as CSMA/CA and AEE-MAC [32]. When the packet size is small, it behaves like a contention-based protocol, whereas with larger packets, it resembles a slot-allocation-based scheme, achieving higher data rates, lower collision rates, and lower energy consumption per transmitted byte.

Table 1 summarizes the most important characteristics of the works described above. From the works described in this section, we can observe that except for Yang et al. [22], the rest of the proposals use, in their respective contention stages, variants of CSMA/CA and ALOHA protocols, which exhibit adequate performance when the number of simultaneous contending users and the overall traffic load are both low. However, they are expected to suffer from congestion as the traffic load and the number of devices increases [6]. Even in the work by Yang et al., where the length of contention period is optimal for each cycle, the random selection of slots by the contending devices could lead to a high number of collisions in a multiple access scenario.

In contrast to the previously described works, in this paper we present a hybrid MAC protocol that employs a different approach during the contention stage. This strategy is described in the following section.

## 3. The 2C Algorithm and Its Adaptable Version

The proposed protocol is based on the 2C algorithm. In this section, we describe how it and its adaptive version perform conflict resolution.

### 3.1. The 2C Algorithm

The 2C algorithm [16] resolves access conflicts by means of random access to a time-slotted channel. There is a central station in charge of examining each time slot with the aim of informing the events that it detected. This latter action is performed through feedback messages that are broadcast to the stations in the network at the end of each slot. In turn, contending stations can be in one of the following two states: transmission ($T_{x}$) and waiting (W). A station with a pending packet has to put itself in state $T_{x}$, and it must wait for the beginning of a time slot to attempt channel access. If only one station transmits, it will be successful and the central station will return an NC (no collision) message. However, if two or more stations transmit, a collision will be detected by the central station and it will return a C (collision) message. In such a case, a collision resolution interval (CRI) will start in the following slot. This interval will finish when all stations involved in the initial collision successfully transmit their packets. A station that generates a transmission request when a CRI is in progress has to wait until it ends.

Let us consider the list symbols shown in Table 2. Each station participating in a CRI maintains a state variable that controls its channel access by switching between $T_{x}$ and W. Let us denote by $r_{t}$ the value of this variable at the beginning of an arbitrary slot *t*. Let $f_{t}$ be the feedback message corresponding to the transmission in slot *t* and received at the end of the same time slot. If the transmission ended in collision, then $f_{t}=C$, otherwise $f_{t}=\mathrm{NC}$.

Let us assume that up to the current slot, all packets have been transmitted (i.e., there is no CRI in progress). Stations with packets to transmit set their state variables to $T_{x}$ and attempt channel access in the following slot. Depending on the sequence of feedback messages received from the central station, each station updates its state variable as follows:If $r_{t}=W$ and $f_{t}=\mathrm{NC}$, then $r_{t+1}=T_{x}$.If $r_{t}=W$ and $f_{t}=C$, then $r_{t+1}=W$.If $r_{t}=T_{x}$ and $f_{t}=C$, then $r_{t+1}=T_{x}$ with probability p=1/2 or $r_{t+1}=W$, also with probability 1/2 .

In the following slot, each station will behave according to the updated value of its state variable. This is repeated until all stations are able to transmit their packets. Note that as collisions occur, more and more stations leave the transmitting state and join the group of waiting stations. This procedure continues until only one station is left in the transmitting state and achieves a successful transmission. In case a successful transmission occurs or all stations abandon the transmitting state, the central station transmits an NC feedback message. Furthermore, note that following an NC feedback message, all stations with $r_{t}=W$ will change their state to $T_{x}$ and will attempt to transmit in the following slot, causing a new collision. Therefore, two consecutive NC feedback messages can only occur at the end of a CRI. This algorithm is called 2C because the contending stations may be either transmitting or waiting, and this situation can be depicted using a two-cell stack. One cell is assumed to contain the group of stations in the transmitting state (i.e., $r_{t}=T_{x}$) and the second cell contains the ones that have deferred transmission (i.e., $r_{t}=W$).

In what follows, we illustrate the operation of the 2C algorithm. As an example, let us consider the wireless network shown in Figure 1, which consists of seven stations (i.e., A–G) and a central station. At a certain point in time, four of them (i.e., A, B, C, and D) have packets to transmit so that they wait for the next CRI to contend. A possible sequence of events is depicted in Figure 2 and is explained below:First slot. At the beginning of the first slot all four stations set their state variables to $T_{x}$ (i.e., $r_{1}=T_{x}$) and transmit. As a consequence, a collision occurs and the central station returns the message $f_{1}=C$.Second slot. In response to the feedback message, the contending stations update their state variables by following the rules mentioned above. That is, they remain in transmission with probability p=1/2 or join the waiting group (also with probability 1/2). In the example, stations A and C choose the former (i.e., $r_{2}=T_{x}$) and B and D the latter (i.e., $r_{2}=W$). Since now two stations transmit, another collision occurs and the central station returns the message $f_{2}=C$.Third slot. Stations A and C react to the feedback message and decide at random whether to continue transmitting or not (both possibilities occur with probability 1/2). In the example, station A moves to the waiting cell (i.e., $r_{3}=W$). In contrast, station C remains in the transmission cell; therefore, $r_{3}=T_{x}$. Since in this slot only station C transmits, a successful transmission occurs, and the central station returns the feedback $f_{3}=\mathrm{NC}$. This event signals the end of a time interval that started with a collision of multiplicity 4 in the first slot and ended when a successful transmission was achieved. This is the concept of what is considered to be a phase in this work (note that the word “phase” has a particular meaning here and, most likely, it differs from other works). A phase begins with either the initial collision of a CRI or the collision that follows after a successful transmission. For identification purposes, let us consider that phase *N* starts with a collision of multiplicity *N*. In all cases, a phase ends when there is a slot containing a successful transmission. In this example, phase 4 starts in slot 1 and ends in the third slot.Remaining slots. When the stations in the waiting cell (i.e., A, B, and D) receive the feedback $f_{3}=\mathrm{NC}$, they switch their state from waiting to transmission in the fourth slot. Therefore, when they transmit again, a new collision occurs. This event starts phase 3. The procedure described above is repeated in each one of the following phases until the last station involved in the initial collision of the CRI successfully transmits. Note that the last transmissions cause two consecutive NC feedback messages, which signals the end of the CRI.

### 3.2. The Adaptive-2C Algorithm

In the original 2C algorithm, when the number of contending stations is large, the resulting CRI length may become relatively long, which can be considered an important drawback. This is due to the fact that the probability of remaining in the transmitting state after a collision is a fixed value (i.e., probability p=1/2) regardless of the number of contending stations. The Adaptive-2C algorithm that we introduced in [18] proposes that *p* should be a function of the number of contending stations *N* (i.e., p=p(N)). For instance, when *N* is large, *p* should be small so that most of these stations should switch to the waiting state in one step of the algorithm. However, it cannot be made too small or the system may transit to a state where none out of the *N* active stations remain in transmission, thus leading to empty slots and waste of resources. In the optimal case, after a collision of multiplicity *N*, only one station should remain in transmission. To this end, in [18] we deduced some alternatives to achieve this goal and found that the following way of computing p(N) yields good results:(1)$$p=p\left(N\right)=\left\{\begin{array}{cc} \frac{1}{2}, & \mathrm{if}\;N=2,\\ \frac{\sqrt{2N(N-1)}-2}{\left(N-2)(N+1\right)}, & \mathrm{if}\;N>\;2. \end{array}\right.$$

In this way, Adaptive-2C achieves a faster collision resolution than the 2C algorithm. The interested reader is referred to the work reported in [18] where a thorough comparison between 2C and Adaptive-2C is reported.

In order to make use of (Equation 1), the central station must estimate the collision multiplicity *N*, and this value should be communicated to all stations in the network at appropriate points in time. Although the estimation of a collision multiplicity can be a difficult task to tackle, in [18] we proposed the following strategy to perform such estimation, which we termed the “update-per-phase policy”. Note that even if it is not possible to know how many stations collided at the beginning of a CRI, this information can be known a posteriori, when such a CRI ends. This is due to the fact that the number of stations involved in the first collision of a CRI equals the total number of successful transmissions observed during the evolution of such a CRI. In this way, the central station can keep track of how many stations contended in the last few CRIs, and it can use this information to estimate the number of contending stations in the following CRI. To this end, several estimation methods can be applied, such as linear prediction.

In the update-per-phase policy, when the first collision of a CRI happens, the central station applies a prediction method to estimate the multiplicity of such a collision (let us denote it by $\hat{N}$), and it provides this value as a feedback message to the network. This information is used by the network stations in order to compute the appropriate value of *p* (by using (Equation 1)), that is, $p=p(\hat{N})$. If a second collision is observed, the stations will set the value of *p* to a fixed value of 1/2 and the system will evolve according to the rules of the 2C algorithm until a successful transmission is achieved. Recall that such success is communicated by the central station by broadcasting an NC feedback message, and this event signals the end of phase *N* and the beginning of phase (N−1). This basic procedure is repeated in each one of the following phases. Note that the central station has to communicate an estimate for the collision multiplicity only once (i.e., at the beginning of a CRI) since every time there is a successful transmission, this estimate can be locally updated at the stations by subtracting 1 from its current value.

The procedure described above constitutes the basis of the update-per-phase policy; however, a couple of comments are in order:Let us assume there is a CRI in progress. Since switching from transmitting to waiting is controlled by a probability, an empty slot may occur because all contending stations may have decided to switch to the waiting state. As it will be explained in an upcoming example, it is necessary to distinguish between an empty slot and a successful transmission. Therefore, instead of using an NC message for both cases, in the case of an empty slot, the central station will broadcast an E (empty) message, forcing all stations to switch back to the transmit state, and a collision will happen. Since all active stations will be involved in such a collision, this situation is identical to the one at the beginning of the current phase, and it will be treated as such. That is, after such a collision, they will use the adaptive value of *p* (i.e., $p=p(\hat{N})$ computed by using (Equation 1)), and afterwards they will use p=1/2 if needed. This is a consideration that was tacitly assumed in [18], but due to its importance for achieving high performance, we emphasize it here.At the system start, there is not enough history to be used in order to compute an estimate for the number of contending stations at the beginning of a CRI. Therefore, depending on the predictor order, a certain number of CRIs have to be completed in order to generate enough data to be used in the prediction. In the meantime, the system has to evolve using the rules of the standard 2C algorithm (i.e., p=1/2 in all cases).

Figure 3 shows a possible evolution of a CRI where channel access is controlled by the Adaptive-2C protocol with updates per phase. This case is similar to the one previously described, that is, at the beginning of a CRI, four out of seven stations contend for the transmission medium (see Figure 1). Let us assume that the system has been in operation for some time so that a prediction algorithm has enough data to operate. The following description illustrates several cases of interest, which are included in Figure 3 (the reader is referred to Table 2 for a list of symbols).

First slot. In the example, stations A, B, C, and D simultaneously transmit, and by using a prediction method, the central station estimates the number of stations that collided (i.e., $\hat{N}$). This value is returned as its feedback message at the end of the first slot. Ideally, $\hat{N}$ should equal 4 in this case.Second slot. With the received feedback, the contending stations calculate $p=p(\hat{N})$. According to this probability, only one of them (i.e., C) remains in transmission, whereas the rest move to the waiting cell. Therefore, station C successfully transmits and the central station returns an NC message at the end of this slot. This event marks the end of phase 4 (note that it started with a collision of multiplicity 4 and ended with a successful transmission). It can be seen that at the end of this slot, the estimated number of stations that remain in contention can be updated as $\hat{N}-1$.Third slot. All stations in the waiting cell move to the transmitting cell so that a collision is observed in the medium and the central station returns a C message.Fourth slot. Each station uses its local estimate for the number of stations that remain in contention (i.e., $\hat{N}-1$) and computes the probability of continuing in the transmitting state (i.e., $p=p(\hat{N}-1)$). In the example, only one of the stations (i.e., D) remains in transmission, and the rest of them move to the waiting cell. Therefore, a second successful transmission is achieved and the central station returns an NC feedback message. Such an event signals the end of phase 3. At the end of this slot, the estimated number of stations in contention becomes $\hat{N}-2$.Fifth slot. The stations in the waiting cell (i.e., A and B) move to the transmitting cell, causing another collision. The central station returns a C message.Sixth slot. By using $p=p(\hat{N}-2)$, the stations decide whether to continue transmitting or not. By chance, all of them decide to move to the waiting cell; thus, an empty slot is observed.Seventh slot. According to the rules of the Adaptive-2C protocol, after an empty slot, all stations in the waiting cell switch their state and transmit. Since another collision is observed, the central station returns a C message.Eighth slot. Note that the collision contained in the seventh slot involved all active stations. However, it came after an empty slot and not after a successful transmission. Therefore, the estimate for the number of contending stations should not be updated with respect to the value used at the beginning of the current phase. Therefore, the stations use the last known value of *p* (i.e, $p=p(\hat{N}-2)$) to determine whether to continue transmitting or not. In the example, station B continues transmitting and A goes to the waiting cell. A successful transmission is observed.Ninth slot. The process continues according to the rules of the algorithm, and the last contending station successfully transmits its data packet. When the corresponding feedback returns, the CRI ends. Note that this event is signaled with two consecutive NC messages, as was also the case in the standard 2C algorithm.

Note that during the evolution of this CRI, there were successful transmissions in slots 2, 4, 8, and 9. Therefore, at the end of the CRI, it is known that the initial collision (in the first slot) had a multiplicity of 4. This information helps the central station to estimate the initial collision multiplicity in the following CRIs.

As a second example, Figure 4, illustrates the different events that can be contained in a time slot and how the transmission probability accordingly changes during the evolution of a CRI. Slot 1 contains the first collision. Slot 2 corresponds to a case where all stations abandon the transmission state and an empty slot occurs. Slot 3 contains a collision after an empty slot. Slot 4 contains a collision that is neither the first collision of the CRI nor a collision after an empty slot. Slots 5 and 6 contain the two successful transmissions.

## 4. The 2CA-R^2^ Hybrid MAC Protocol

In this work, we introduce 2CA-R^2^, which is a hybrid MAC protocol consisting of two operational stages: a CRI and a DTI, or collision resolution interval. During the CRI, all active stations (i.e., stations with a data packet to transmit) contend to reserve a contention-free time slot. Such reservations are used during the DTI, which comes after the CRI. In the DTI, the stations take turns to transmit, and the order in which they access the channel corresponds to the same order in which they achieved their reservation in the CRI. The distinguishing feature of 2CA-R^2^ is that contention in the CRI is resolved using the Adaptive-2C algorithm with the update-per-phase policy.

After the brief overview provided above, the following three sections describe in more detail the 2CA-R^2^ protocol. For a list of symbols, the reader is referred to Table 2 above.

### 4.1. Collision Resolution Interval

In the CRI, time is slotted using short-length slots or “minislots”. The central station is in charge of examining all minislots and providing appropriate feedback depending on the detected event. This feedback can be one of the following four messages: $\hat{N}$ (estimated value of *N*), C (collision), E (empty), and NC (no collision). Let us denote with $f_{t}$ the feedback message corresponding to minislot *t*.

During a CRI, each station keeps a state variable $r_{t}$ that can take on the following values: $T_{x}$ and W; further, it is updated according to the feedback messages. Each station also makes use of variable *s* to store its reserved transmission slot and variable *N* to keep track of the number of stations that are estimated to remain in contention.

When a station has a packet to transmit, it sets its state variable $r_{t}$ to $T_{x}$, meaning that it must wait for the next CRI. As a consequence, a station that became active after the start of a CRI must wait until the central station announces the start of a new CRI. Such a station also sets *s* to 0 and *N* to 1.

The beginning of a CRI is signaled by the central station with a RESERVATION_BEGIN broadcast message. A station with $r_{t}$ set to $T_{x}$ will send an RREQ in the first minislot of the CRI. The RREQ will be successful only if $r_{t}=T_{x}$ and $f_{t}=\mathrm{NC}$. The CRI will last only one minislot if only one station sends an RREQ (i.e., this is equivalent to immediate random access). However, if two or more stations transmit, a collision will occur and a conflict resolution procedure will begin (i.e., controlled access is enforced). The CRI evolves according to the following rules:If ($r_{t}=T_{x}$ and $f_{t}=\mathrm{NC}$), then s=s+1. This rule corresponds to a case where a station transmitted a reservation request which was successful. This rule implies that such a station stores its reservation position in variable *s* and leaves contention. Variable *s* will not be changed any further during the remainder of the contention period.If ($r_{t}=T_{x}$ and $f_{t}=\hat{N}$), then $N=\hat{N}$. This rule is triggered when a station transmitted in minislot *t* and the first collision of a CRI occurs. In this case, the central station returns the estimated value of the collision multiplicity (i.e., $\hat{N}$), which is copied into the local variable *N*, and this value is used to compute probability p=p(N). In turn, $r_{t+1}$ maintains the value $T_{x}$ with probability *p* or $r_{t+1}=W$ with probability 1−p.If ($r_{t}=T_{x}$ and $f_{t}=C$), then $r_{t+1}$ maintains the value $T_{x}$ with probability *p* or $r_{t+1}=W$ with probability 1−p. This rule determines what to do when a station transmitted and a collision is detected in the medium. The following two cases are possible: if *t* is either the first minislot of a phase or the one that follows after an E feedback message, the station uses the current value of *N* to calculate p=p(N). Otherwise, the station uses p=0.5.If ($r_{t}=W$ and $f_{t}=E$), then $r_{t+1}=T_{x}$. The station was in the waiting group and no one transmitted. Therefore, it sets its state to $T_{x}$ in order to transmit in the following minislot.If ($r_{t}=W$ and $f_{t}=\mathrm{NC}$), then $r_{t+1}=T_{x}$, s=s+1 and N=N−1. The station was in the waiting group and someone else achieved a successful reservation. Therefore, it enters contention, increases the reservation position, and decreases by one the number of contending stations.If ($r_{t}=W$ and $f_{t}=C$), then $r_{t+1}$ remains in state W. The station was in the waiting group and there was a collision in the medium. Therefore, it remains in the waiting state.

The CRI finishes when two consecutive NC feedback messages occur (note that this situation can only happen at the end of a CRI). As a result of this algorithm, at the end of the CRI, each station knows its position within the transmission frame.

### 4.2. Data Transmission Interval

Once the CRI finishes, the DTI starts. Stations transmit their packet, taking turns according to the order in which they achieved their reservation. The end of the DTI is implicitly notified by the central station when it broadcasts the RESERVATION_BEGIN message and a new cycle starts.

### 4.3. Example

An example of the operation of 2CA-R^2^ is shown in Figure 5. The network consists of seven stations A–G and the central station (see Figure 5c). For the sake of simplicity, let us consider a CRI that evolves in the same way as the example explained in the previous section. That is, initially only four stations A–D have packets to transmit and send an RREQ message in the first minislot of the CRI. The resulting collision will be solved by the rules of the Adaptive-2C conflict resolution algorithm, leading to the sequence of events depicted in Figure 5a. Note that the stations C, D, B, and A achieved a successful transmission in slots 2, 4, 8, and 9, respectively. Therefore, the former sequence also becomes the order in which they transmit their data packets during the DTI (see Figure 5b).

## 5. Modeling the 2CA-R^2^ Mechanism as a Markov Chain

The Markov chain model introduced in this section represents the time evolution of the contention interval of the 2CA-R^2^ protocol, which is based on Adaptive-2C with updates per phase. Each state of the chain is identified by the ordered pair (TC, WC), where TC and WC indicate the number of packets in the transmission cell and in the waiting cell, respectively. Figure 6 depicts the model where the states of the chain are represented as circles.

The “Start” labels in Figure 6 indicate the possible starting states of a CRI depending on the number of stations that collided in its first minislot. Let us also recall that a collision involving all currently active stations marks the beginning of a phase. Therefore, phase *n* starts with a collision of multiplicity *n*, i.e., in state (n,0), and ends with a successful transmission in state (1,n−1). When phase *n* ends, the system transitions to phase n−1, where there are n−1 contending stations.

Let A=(a,b) and B=(c,d) be two states of the chain. Then, the probability of transitioning from the former to the latter can be written as $P_{A,B}$ or, more explicitly, as $P_{\left(a,b),(c,d\right)}$. However, based on the following observation and for readability reasons, we use a simplified notation for transitions among states that belong to the same phase. Let us note that in any given phase *n*, the following observation holds. If there are *i* stations in the transmission cell (with *i* in the interval 0≤i≤n), then there must be n−i stations in the waiting cell. Therefore, a state in phase *n* can be fully identified by using only one subscript (i.e., *i* in this case). Therefore, let us denote a transition from state *i* to state *j* of phase *n* by $P_{i,j}^{\left(n\right)}$. Figure 6 shows the transition probabilities as labels placed along the arrows that connect the states. Table 3 summarizes the notation used in the model description.

In our analysis, it is convenient to distinguish among three types of transitions. A type 0 transition occurs when it departs from the initial state of a phase. In turn, a type 1 transition is originated at a non-initial state. Finally, a type 2 transition is the one that occurs with probability 1. This last situation occurs in the following two cases: (a) when state (0,n) is reached (empty minislot), the only possibility is to transition to the initial state of the same phase (i.e., (n,0)); and (b) when state (1,n−1) is reached (successful transmission), the only possibility is to transition to the initial state of the following phase, that is, (n−1,0).

As an example of the transition types, Figure 7a shows all transitions corresponding to phase n=3. The following three types of transitions can be identified:Type 0 transitions. Figure 7b shows all possible transitions from the initial state (3,0) (lettering in blue).Type 1 transitions. Figure 7c shows all possible transitions departing from (2,1), which is not an initial state (lettering in purple).Type 2 transitions. Figure 7d shows the cases with only one possible destination (lettering in red).

In summary, the type of transition depends on the kind of state from which it departs. Therefore, it is convenient to observe that a state that belongs to phase *n* can be classified into one out of the following three possibilities: (a) the initial state, i.e., (n,0); (b) non-initial states, which belong to the set {(i,n−i)|2≤i≤n−1}; and (c) states with only one possible destination, i.e., (0,n) and (1,0). We take into consideration this classification in order to compute the probability of transitions that depart from a state in phase *n* (with n≥2), as follows:Type 0 transitions, from the initial state (n,0) to (n−j,j):
(2)Pn,n−j(n)=nn−jp(n−j)(1−p)j,
where$$\begin{array}{cc} \; & \;j=0,\ldots ,n;\;p=\left\{\begin{array}{cc} \frac{1}{2} & \;\mathrm{if}\;n=2,\\ \frac{\sqrt{2n(n-1)}-2}{\left(n-2)(n+1\right)} & \;\mathrm{if}\;n>2. \end{array}\right. \end{array}$$Type 1 transitions, from a non-initial state (i,n−i) to (j,n−j):
(3)Pi,j(n)=ijpj(1−p)(i−j),
where$$\begin{array}{cc} \; & 2\leq i\leq n-1;\;j=0,\ldots ,i;\;p=\frac{1}{2}. \end{array}$$Type 2 transition, from (0,n) to (n,0) (after an empty minislot, all stations in the waiting state switch their state to transmission):(4)$$P_{0,n}^{\left(n\right)}=1.$$Type 2 transition, from (1,n−1) to (n−1,0) (after a successful transmission, the CRI evolves from phase *n* to phase n−1).(5)$$P_{\left(1,n-1),(n-1,0\right)}=1.$$Note that in this case, the simplified notation cannot be used since the origin and destination belong to different phases.

Note that a type 0 transition (i.e., the one departing from the initial state of a phase) is the only type of transition using a value of p=p(N) (calculated by (Equation 1)) since it is only at the beginning of a phase that a station knows the estimated number of contending stations *N* and can use that value to optimize the value of *p*. It is worth remembering that stations can distinguish the beginning of a phase only when they receive a successful transmission feedback message or when they receive an empty minislot message.

The last phase of the Markov chain starts when the system reaches state (1,0) and the only remaining station is able to transmit without contention. In this case, the system transitions to state (0,0) and the CRI ends. For this reason, this state is considered to be an absorbing state in the chain.

The transition probabilities described above are used to create transition matrix P, which represents an entire CRI. Figure 8 illustrates the general form of this matrix, and Figure 9 shows a numerical example for a CRI that starts with a collision multiplicity of n=4.

With the information in transition matrix P, a number of metrics of performance can be computed. For our purposes, it is of particular interest to calculate the average CRI duration. To this end, we can apply some known results from the theory on Markov chains and compute the “hitting” times [33] by solving the following system of linear equations:(6)$$H_{A,B}=\left\{\begin{array}{c} 0\;\mathrm{for}\;A=B,\\ 1+\sum_{I\neq B}P_{A,I}H_{I,B}\;\mathrm{for}\;A\neq B, \end{array}\right.$$
where *A*, *B*, and *I* are three states of the chain and $H_{A,B}$ is the expected number of time minislots required to reach state *B* departing from state *A*. From the set of results provided by solving (Equation 6), we are interested in $H_{A,B}$ where A=(n,0) and B=(0,0) for different values of *n*. This is due to the fact that a CRI starts with a collision of multiplicity *n* (i.e., at state (n,0) in phase *n*) and ends when the system reaches the absorbing state (i.e., state (0,0)).

In this way, we computed the average number of minislots ($ms_{n}$) that it takes to resolve an initial collision of size *n*, and the results are shown in Table 4. For comparison purposes, in our tests we used computer simulations and two different software tools to solve the system of linear equations implied by (Equation 6). These results are explained below:The first column corresponds to the results obtained via computer simulations. To this end, we used the OMNeT++ v5.7 simulator [34], which is a framework for modular object-oriented discrete-event network simulation. Each value shown in this column comes from averaging 1000 simulation runs.The second column shows the results obtained from a Matlab R2024a script that finds the minimum norm non-negative solution of the system of linear equations [35]. The case indicated as NA (not available), for n=256, arose from software limitations regarding the use of matrices.The last column shows results obtained with a Python v3.11 script that applies the theory of Markov chains to compute the mean time before absorption [36].

As can be seen, the number of minislots $ms_{n}$ needed to resolve a collision of size *n* obtained by simulating the 2CA-R^2^ with the update-per-phase simulation model in OMNeT++ is practically the same as that obtained using the Markov chain, which indicates that the mathematical model presented here accurately represents the simulation model used under ideal conditions (i.e., error-free channel). It is worth pointing out that more detailed models can be derived for this system. For instance, it does not consider the effect of channel errors on the MAC layer. However, as has been shown in this section, the model herein introduced represents a good trade-off between high accuracy and reasonable complexity.

From the values shown in Table 4, the mean throughput in Mbps can be easily computed if packet sizes and the data rate are known. For instance, it can be computed by(7)$$Th_{n}=\left(\frac{\left(65\right)\left(8\right)\left(n\right)\left(1\times 10^{-6}\right)}{\left(ms_{n}\right)\left(1.68\times 10^{-4}\right)+\left(n\right)\left(5.28\times 10^{-4}\right)}\right),$$
where $1.68\times 10^{-4}$ is the time (in seconds) required to send an RREQ of 20 bytes at 1 Mbps (i.e., the time duration of a minislot) and receive a feedback message of 1 byte from the central station. In turn, $5.28\times 10^{-4}$ is the time (in seconds) required to send and acknowledge a data packet with a payload of 65 bytes at 1 Mbps. This time is the sum of the transmission time plus the time required to receive the corresponding feedback message of 1 byte from the central station.

Figure 10 shows a comparison between the results obtained with (Equation 7) and mean values obtained from 1000 OMNeT simulations under the same conditions. It is worth mentioning that both curves are indistinguishable from each other because they overlap. This result verifies the validity of the developed mathematical model. Since the Markov model assumes an error-free channel, for comparison purposes we are also including results considering 1% and 5% packet error rates (PERs). Such results show the effect of channel errors on the throughput forecasted by the model. All experimental results are presented using confidence intervals with a 95% confidence level.

## 6. Simulation Experiments and Results

We evaluated the performance of the proposed protocol by means of discrete event simulations. To this end, we implemented the following simulation models in OMNeT++ version 5.7. We developed a simulator of 2CA-R^2^, and for comparison purposes, we also implemented a variant of the protocol where access conflicts are resolved by means of the standard 2C algorithm instead of Adaptive-2C. This second set of results is identified in the graphs with the label “2C-R^2^”, but these results are included only where we deemed illustrative to show a comparison between the two approaches. In addition, to assess the performance regarding random access, we also made use of the OMNet model for DCF, the fundamental MAC technique of the IEEE 802.11ah standard [37], which is one of the main candidate technologies to support IoT applications in WLANs in the coming years [38]. The implementation follows the standard behavior described in the 802.11 specification (including Retry Limit and the RTS/CTS mechanism). In this section, we report our findings.

### 6.1. Metrics of Performance

In this section, we define the metrics of performance to be used in the evaluation. Note that by central station we refer to an access point, which is the common term used in Wi-Fi.

Network throughput ($T_{N}$). It is the total number of bits received at the central station (#bits) divided by the simulation time ($t_{s}$). That is,(8)$$T_{N}=\frac{\#bits}{t_{s}}.$$Access delay ($D_{A}$). This is the time that elapses from the instant a packet is generated at a network node ($t_{g}$) to the instant the corresponding acknowledgement (returned by the central station) is received ($t_{r}$).(9)$$D_{A}=t_{r}-t_{g}.$$Queuing delay ($D_{Q}$). This is the time that elapses from the moment a packet is generated at a node of the network ($t_{g}$) until the beginning of its first transmission attempt ($t_{a}$) (that is, until there is no other packet ahead of it in the queue).(10)$$D_{Q}=t_{a}-t_{g}.$$Percentage of dropped packets ($\%p_{d}$). It is computed as the number of dropped packets ($\#p_{d}$) divided by the total number of packets transmitted in each simulation ($\#p_{Tx}$). That is,(11)$$\%p_{d}=\frac{\#p_{d}}{\#p_{Tx}}.$$

### 6.2. Simulation Scenarios

The simulation scenario considers a centrally located access point and a fixed number of uniformly distributed nodes or stations. To observe the protocol performance under different network loads, we considered a number of stations that was varied from 1 to 256. We also considered that each one of them behaves as a “greedy” source, which leads to what is frequently known as saturation conditions. That is, as soon as a node successfully sends a data packet, it generates another one to be transmitted. All simulations were run with a data rate of 1 Mbps and a packet payload of 65 bytes. These values of data rate and packet size are commonly found in different types of IoT applications [39,40]. For the chosen data rate and packet size, we configured each simulation scenario with a value of bit error rate (BER) so that it produced 1% packet loss. When a packet is lost, the AP emits an error message and the affected packet is retransmitted in the following CRI. In addition, for comparison purposes, the RREQ size was chosen to be 20 bytes, which is the same length used by DCF for the RTS messages. However, the RREQ size can be decreased, since all that is required is the ID of the transmitting station. For 2CA-R^2^, the predictor order to estimate $\hat{N}$ was set to 1. Each case was simulated 10 times in order to obtain mean values and provide their confidence intervals with a 95% confidence level. The whole set of parameter values is shown in Table 5.

### 6.3. Network Throughput

We can see in Figure 11 that when there are a few contending stations (between 1 and 20), the performance of 2CA-R^2^ starts with high figures but quickly decreases as the number of stations increases. After a certain point (approximately after 40 stations), its performance nearly does not change as the number of stations increases. Similar general trends are observed for both DCF and 2C-R^2^, but the network throughput achieved by 2CA-R^2^ in all cases is greater than the one achieved by both of them. Furthermore, significant improvements are observed at both ends of the test conditions. With a large number of devices, for instance, the improvement that 2CA-R^2^ achieves over DCF is about 40%.

### 6.4. Access Delay and Packet Loss

We also took measurements of access delay in the previously described simulation conditions. The corresponding results are shown in Figure 12.

It is interesting to note that when the number of contending stations is greater than 150, the figures of access delay are better for DCF than both 2C-R^2^ and 2CA-R^2^. However, this is due to the amount of packets that were dropped by DCF in these cases, when they exceeded the transmission Retry Limit. In contrast, for 2C-R^2^ and 2CA-R^2^, no packets were dropped due to this reason since there was not a Retry Limit in effect for these protocols. For the same simulation scenarios, Figure 13 shows the percentage of dropped packets by DCF, which is the only protocol that drops packets.

We can observe that in the worst scenario, that is, when there were 256 stations simultaneously transmitting, around 16% of their packets were discarded. It is important to note that these packets did not contribute to the access delay statistic since they were never successfully delivered. In this way, the Retry Limit parameter in DCF helps to keep access delay under control. Therefore, the time delay will depend on the value chosen for such a parameter at the cost of discarding some data packets coming from the stations.

The results obtained in this scenario in terms of network throughput and access delay show that the performance of 2C-R^2^ was poor compared to DCF and 2CA-R^2^. For this reason, results from the 2C-R^2^ mechanism are not included in the remaining test scenarios.

### 6.5. Delivery Time of the Complete Set (One Packet per Station)

Additional measurements were taken to compare 2CA-R^2^ and DCF. To this end, another simulation scenario was implemented using the same conditions as the ones used in the previous simulations. The difference was that in this case, each one of the *n* devices generated only one packet, which they tried to transmit at the beginning of the simulation. The metric of performance was the time that the system took to transfer the whole set of *n* packets to the access point.

Regarding the 2CA-R^2^ mechanism, in Figure 14 we can see that after 20 devices, it can deliver the entire set of packets in less time than DCF, reaching a difference of more than 100 ms for 256 devices. As an example of the importance of this result, let us consider a situation where the time to deliver data packets is limited to 250 ms. As depicted in Figure 14 (see the dashed red line), approximately 145 stations can transmit their data packet using DCF, whereas in the same time, approximately 203 stations are able to transmit using 2CA-R^2^. That is, at this point, 2CA-R^2^ achieves an improvement of roughly 40% with respect to DCF. As can be seen in the figure, larger gains are expected as the red line moves further up. This result emphasizes the importance of developing more efficient MAC algorithms able to support the communication needs of the highly populated IoT networks that we envision to appear in the near future.

### 6.6. Experiments with Poisson Traffic

A final set of measurements was taken to compare 2CA-R^2^ and DCF under saturation-free channel conditions. To this end, each station in the simulation generated packets according to a Poisson process with parameter λ=2.5 packets per second. This value was selected to maintain the total network traffic below the channel data rate, even in the case with the largest number of stations in the system. At the same time, this value allowed us to observe the system response under a wide range of conditions of channel utilization when the number of stations in the network was varied. In addition, each station was configured with a maximum queue size of five packets, which proved to be large enough to reduce buffer overflow to negligible levels. In this set of experiments, an error-free channel was assumed.

The performance of the mechanisms was measured as indicated by (Equation 9), that is, in terms of the delay that a packet experiences measured from its generation time to the instant when the transmitter obtains a notification about the successful reception. Note that with greedy traffic generation, this delay is only due to channel access since a new packet is generated only if the previous one has been delivered so that there is no queue buildup at the transmitter. In contrast, with Poisson traffic, the packet generation and transmission are independent processes. Therefore, a new packet can be generated even if the previous one has not been delivered, thus leading to possible queue buildup. Therefore, in this case, part of the access delay is due to queuing delay.

As in the previous simulation experiments, each case was simulated 10 times in order to obtain mean values and provide their confidence intervals with a 95% confidence level.

Figure 15 shows the corresponding results for access delay. It can be seen in the figure that under the described conditions, the performance of the 2CA-R^2^ mechanism is superior to the one achieved by DCF, a situation that becomes more noticeable when the number of stations is increased.

In order to explain this result, it is convenient to consider the queuing delay for each protocol. Figure 16 shows that the queuing delay when using DCF increases as the number of devices increases. In fact, in the worst case, i.e., when there are 256 devices in the scenario, there is on average about 3 ms of queuing delay. In turn, the queuing delay when using the 2CA-R^2^ protocol is very close to 0 regardless of the increase in the number of devices. Thus, it is noticeable that the mean access delay for 2CA-R^2^ is entirely due to the collision resolution time, unlike DCF where this delay has a significant contribution to the mean access delay, a situation that is exacerbated by increasing the number of devices.

It is important to emphasize that under these traffic conditions, not all stations participated in the initial collision of each CRI of 2CA-R^2^. This is due to the fact that the number of active stations in different CRIs varied due to the bursty nature of the source. Thus, one may think that estimating the value of *N* in the same way as before would not be adequate (i.e., making it equal to the number of successful transmissions in the previous CRI). However, Table 6 summarizes some results that can be used to assess the suitability of this strategy. These results are the average fraction of CRIs starting with one, two, and more than two contending devices. The last column in Table 6 shows the channel utilization achieved by the network in each case.

As can be seen, in this scenario, a high percentage of CRIs of the 2CA-R^2^ started with only one device trying to transmit (therefore, they were granted immediate access). This situation prevailed even when the number of devices was increased to large values. The average fraction of CRIs starting with two or more devices was relatively small, and the access conflict was efficiently handled by the MAC protocol so that their impact on the average access delay was negligible. Additionally, in [18], it was shown that the algorithm is robust in the presence of estimation errors in the number of colliding stations. As a consequence, in the case of abrupt changes in the number of stations between consecutive CRIs, a significant performance degradation is not expected.

### 6.7. Discussion on Future Research

A number of potential research avenues can depart from this work. We believe that the following ones are worth considering:Collision estimation. In our implementation of 2CA-R^2^, the previous CRI length is used to estimate the number of contending stations in the following CRI. Although this strategy proved to be effective in our tests, we emphasize that the proposal is not tied to this prediction method, and many others can be employed instead. In this context, we believe that the performance of the protocol can be improved by applying AI techniques. A particularly promising research direction involves the development of a machine learning-based mechanism to predict the collision multiplicity. There exist some previous works in this direction, such as the one reported in [41] for LTE networks.Integration with fog computing. The collision estimation is assumed to take place at the central station, which is considered to be a more powerful machine than the IoT devices. However, if needed, this task can also be offloaded to even more capable machines by using the concept of fog computing, where large datasets can be analyzed to identify traffic patterns that can be used to produce more reliable estimates.Protocol scaling. It is left as future work to test the protocol performance with more than 256 stations and to implement some scaling strategies. The results of this work can be directly applied to a situation where the number of contending stations is larger but is divided into contending groups of up to 256 stations each. This scaling strategy is also used by 802.11ah.Simulation of large-scale IoT networks. An important research direction is related to being able to carry out simulations of large-scale networks. To this end, one promising approach that has been proposed is the use of the agent-based computing paradigm combined with traditional network simulation. The interested reader is referred to the work by Savaglio et al. [42] and references contained therein.

## 7. Conclusions

In this work, we introduced 2CA-R^2^, a hybrid MAC protocol for M2M communications. The proposal consists of a collision resolution interval followed by a data transmission interval. Although this structure is similar to the one used in some other hybrid protocols, what sets it apart is that contention is resolved by means of the Adaptive-2C algorithm, which dynamically adjusts the transmission probability as a function of the estimated number of contending stations.

We modeled the protocol using a Markov chain and evaluated its performance by means of discrete event simulations. As a performance reference, we used DCF, the widely popular MAC technique. With greedy sources, the results showed that 2CA-R^2^ achieves higher levels of network throughput than DCF and lower access delays when there are less than 170 devices. With a larger number of devices, DCF exhibits lower access delays but in exchange for discarding more than 10% of the offered traffic (whereas our proposal was not configured to drop packets). We also measured the time it takes to transfer the data packets coming from a varying number of sources (one packet per source). 2CA-R^2^ was shown to be able to transfer the whole set of packets in significantly shorter times than DCF. Additional simulations with Poisson traffic were performed, and the protocol maintained its advantages.

Under the test conditions used in this evaluation, 2CA-R^2^ exhibited significantly better performance than DCF. These results suggest that the proposed protocol supports the communication requirements of a large number of IoT devices better than what can be achieved with currently popular MAC techniques.

## Figures and Tables

**Figure 1 sensors-25-02994-f001:**
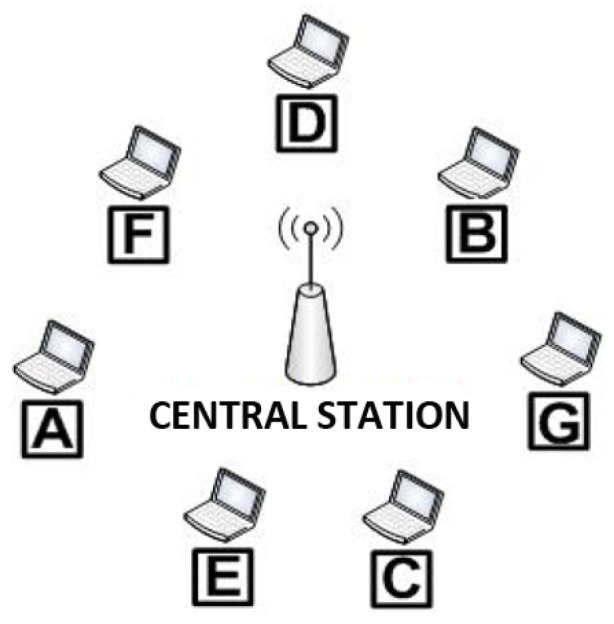
Example of a wireless network.

**Figure 2 sensors-25-02994-f002:**
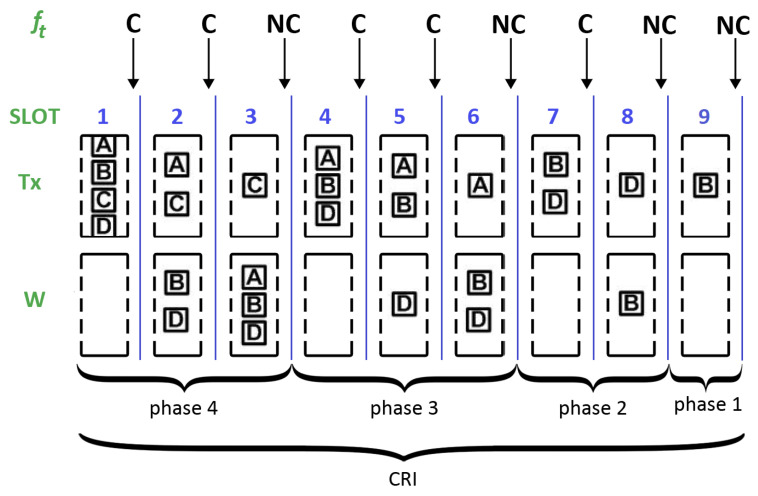
Example of the operation of the 2C algorithm.

**Figure 3 sensors-25-02994-f003:**
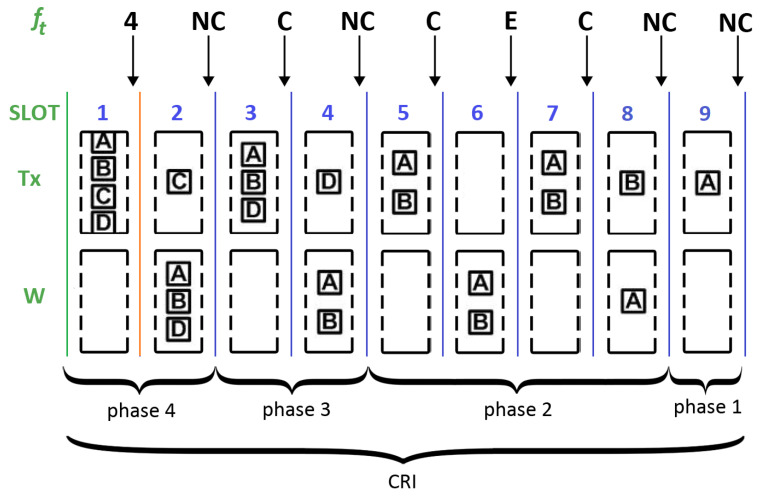
Example of a CRI using Adaptive-2C with the update-per-phase policy.

**Figure 4 sensors-25-02994-f004:**
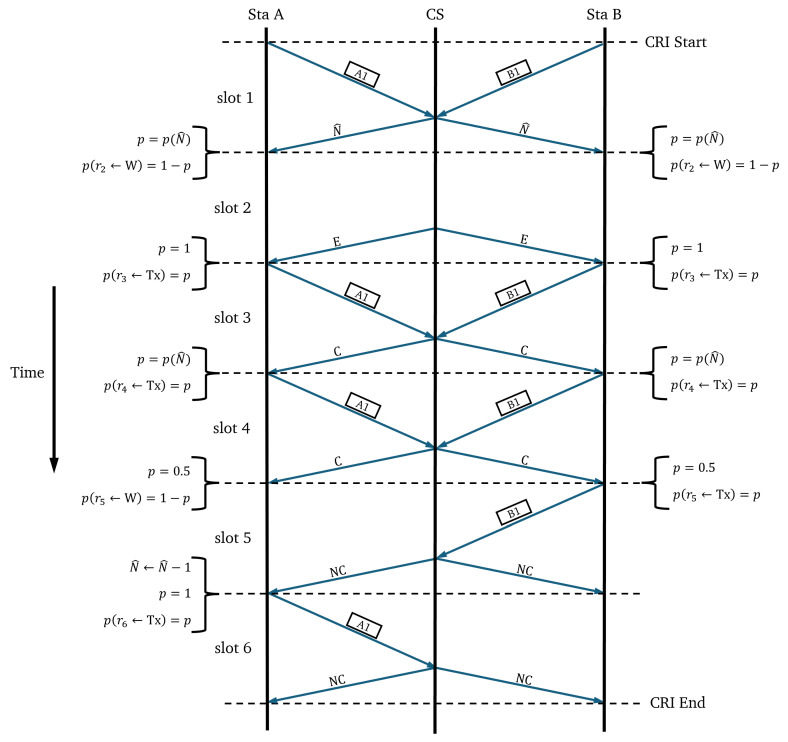
Timing diagram of a CRI corresponding to the Adaptive-2C algorithm in a network consisting of two stations (i.e., “Sta A” and “Sta B”) and a central station (CS).

**Figure 5 sensors-25-02994-f005:**
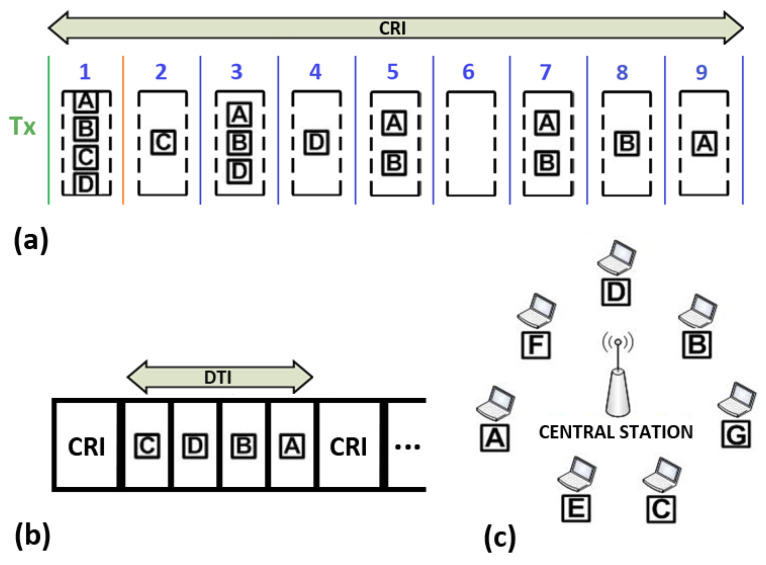
Example of the operation of 2CA-R^2^. (**a**) Evolution of the CRI (transmission cell); (**b**) evolution of the DTI; (**c**) network.

**Figure 6 sensors-25-02994-f006:**
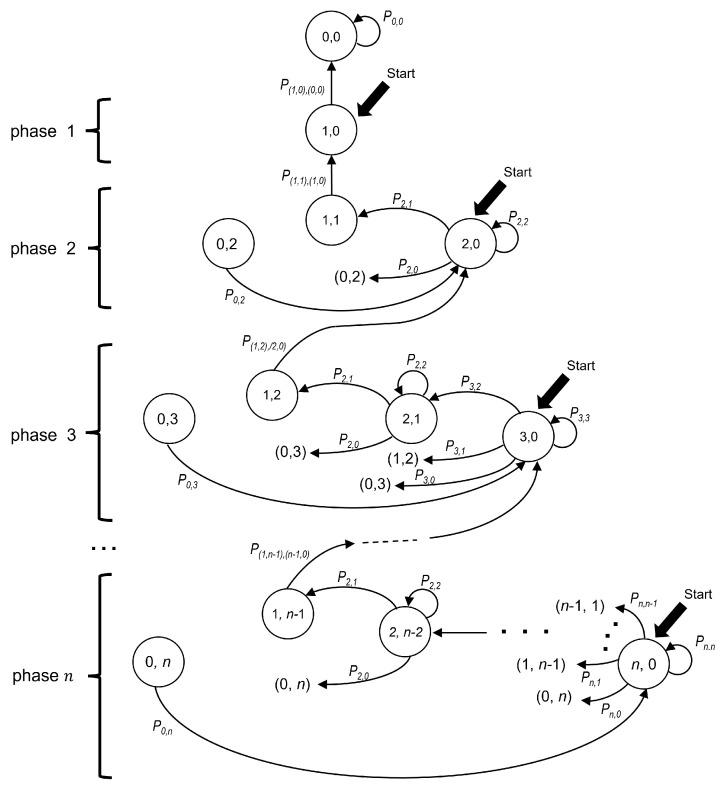
Markov chain model for Adaptive-2C with update per phase. For readability reasons, the superscript indicating the phase of some probabilities has been omitted because this information is shown to the left.

**Figure 7 sensors-25-02994-f007:**
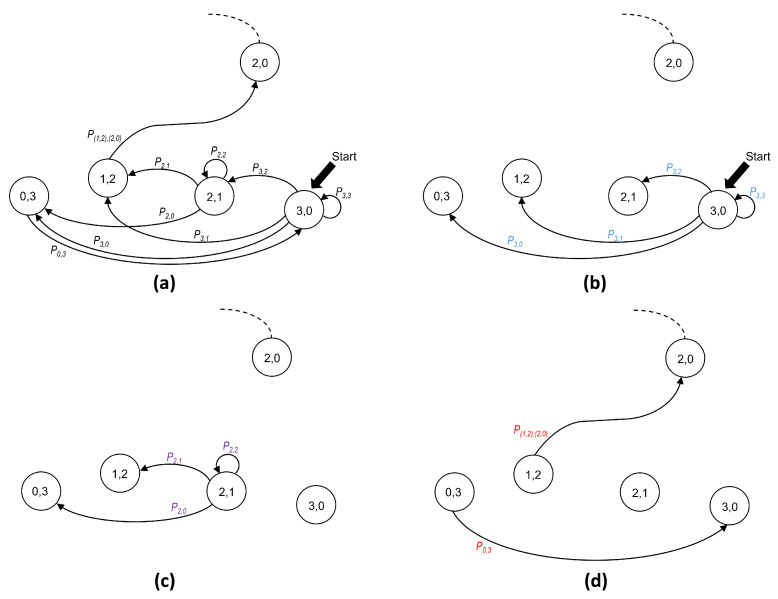
Possible transitions for a CRI in phase n=3. (**a**) All transitions; (**b**) type 0 transitions; (**c**) type 1 transitions; (**d**) type 2 transitions.

**Figure 8 sensors-25-02994-f008:**
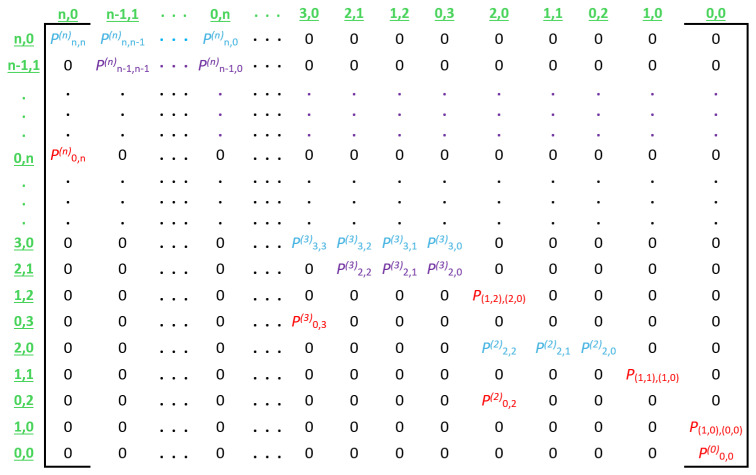
Transition matrix P. For the sake of clarity, in this illustration, each row is labeled in green with the initial state that it represents. In the same way, each column is labeled in green with the corresponding destination state. Furthermore, note that the transition types are distinguished by color. Blue, purple, and red are used for transition types 0, 1, and 2, respectively.

**Figure 9 sensors-25-02994-f009:**
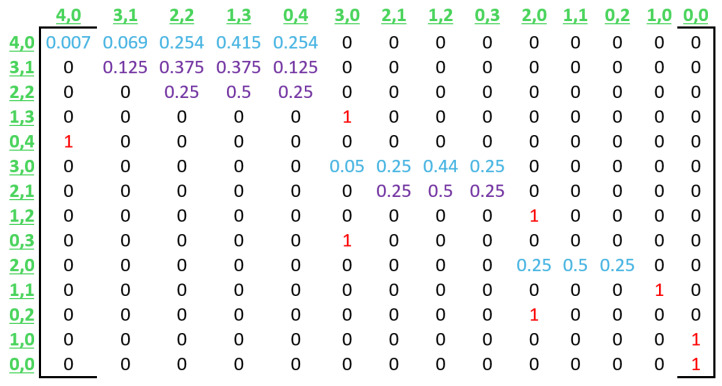
Numerical example of transition matrix P for n=4. The colors have the same meaning as in the general form of the matrix shown in Figure 8.

**Figure 10 sensors-25-02994-f010:**
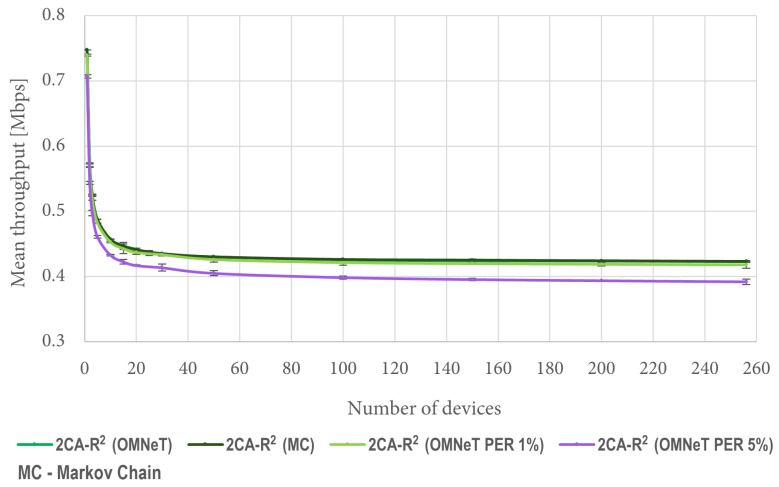
Mean throughput comparison between results from OMNet simulations and the Markov chain model.

**Figure 11 sensors-25-02994-f011:**
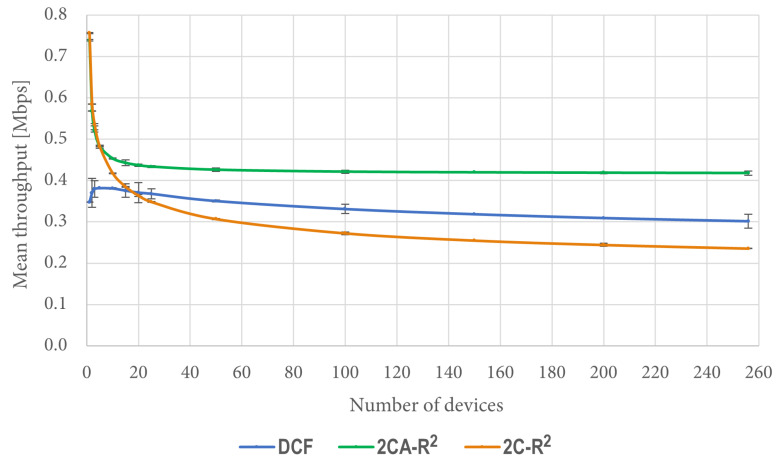
Mean network throughput.

**Figure 12 sensors-25-02994-f012:**
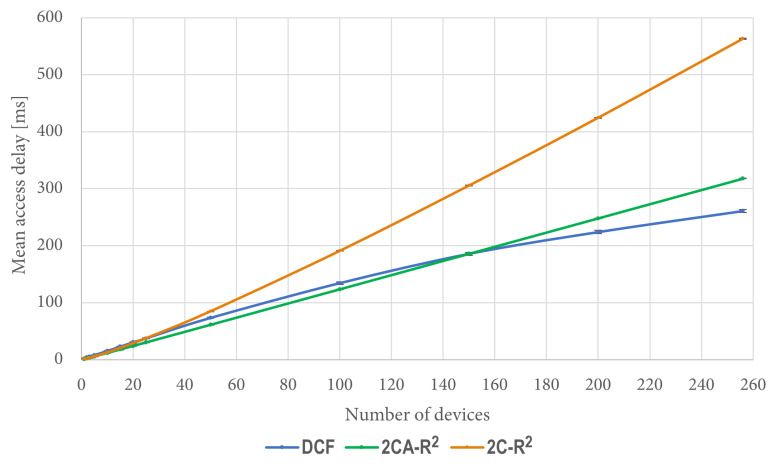
Mean access delay.

**Figure 13 sensors-25-02994-f013:**
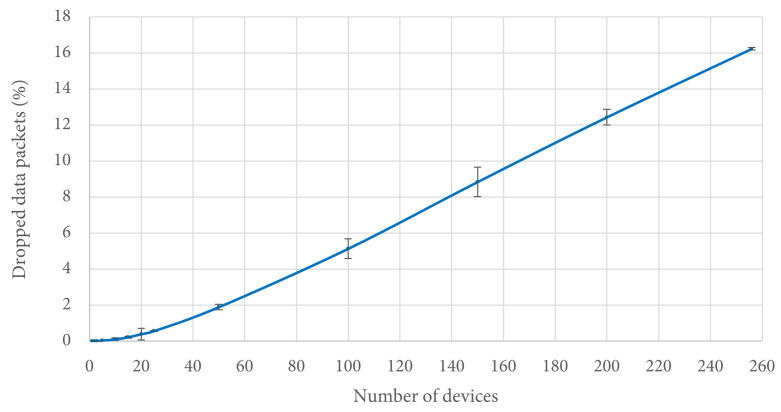
Percentage of dropped data packets.

**Figure 14 sensors-25-02994-f014:**
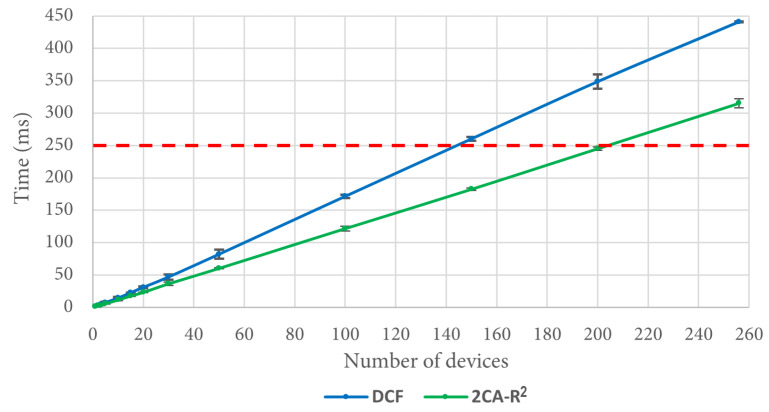
Mean delivery time of a complete set of packets

**Figure 15 sensors-25-02994-f015:**
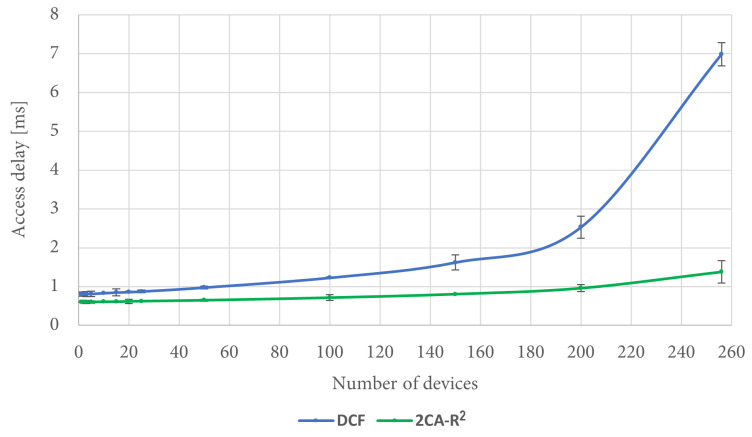
Access delay comparison between 2CA-R^2^ and DCF with Poisson traffic.

**Figure 16 sensors-25-02994-f016:**
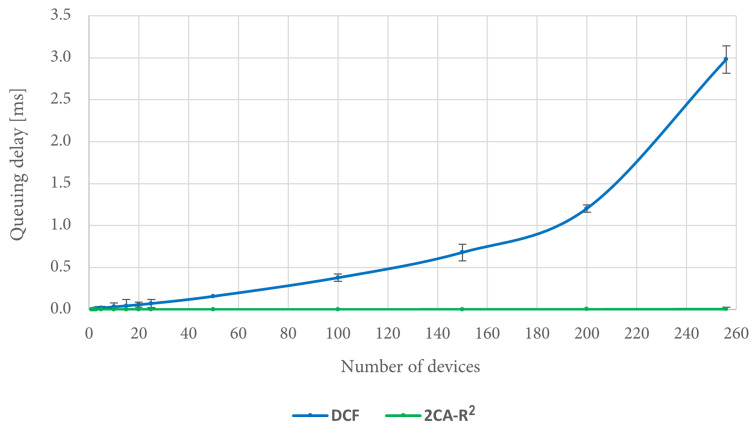
Queuing delay comparison between 2CA-R^2^ and DCF with Poisson traffic.

**Table 1 sensors-25-02994-t001:** Summary of hybrid MAC mechanisms and protocols for machine-to-machine (M2M) communications.

Year	Authors	Reservation Mechanism	Channel Access Method	No. of Devices	Data Rate	Payload Size [bytes]
2013	Y. Liu et al. [19]	*p*-persistent CSMA	TDMA	1–500	1.7 Gbps	1728
2014	P.K. Verma et al. [20]	CSMA	DCF within TDMA slots	1–500	1.5 Gbps	1500
2017	E. Hegazy et al. [21]	NP-CSMA	TDMA	100, 200, 500	NA	NA
2018	B. Yang et al. [22]	Random slot reservation	TDMA	1–300	11 Mbps	1000
2018	W. Saad et al. [25]	s-ALOHA	TDMA	1–1200	256 kbps	16
2020	A. Lachtar et al. [26]	*p*-persistent CSMA	TDMA	500, 800, 1200	1.7 Gbps	1728
2020	D. Olatinwo et al. [27]	s-ALOHA	TDMA	1000	256 kbps	16
2022	D. Olatinwo et al. [28]	CSMA	TDMA	3–15	5 Mbps	NA
2024	X. Fan et al. [31]	CSMA/CA	TDMA (if necessary)	1–100	256 kbps	50–500

NA: not available.

**Table 2 sensors-25-02994-t002:** List of symbols used in the description of 2C, Adaptive-2C. and 2CA-R^2^.

Symbol	Description
$r_{t}$	Station state at the beginning of slot *t*; it can be waiting (W) or transmitting (T_x_)
$f_{t}$	Feedback message returned by the central station at the end of slot *t*
*p*	Transmission probability
p(·)	Adaptive transmission probability
$\hat{N}$	Estimation of the collision multiplicity
*s*	Reserved transmission slot

**Table 3 sensors-25-02994-t003:** List of mathematical symbols used in the Markov chain model of 2CA-R^2^.

Symbol	Description
*n*	Phase number
TC	Number of stations in transmission cell
WC	Number of stations in waiting cell
$P_{i,j}^{\left(n\right)}$	Probability of going from state (i,n−i) to state (j,n−j) in phase *n*
P	Transition matrix
$H_{A,B}$	Expected number of minislots required to reach state *B* departing from state *A*

**Table 4 sensors-25-02994-t004:** Average number of minislots needed to resolve a collision of size *n* using 2CA-R^2^ with updates per phase.

*n*	OMNeT++ $\left[{\mathrm{ms}}_{n}\right]$	Matlab $\left[{\mathrm{ms}}_{n}\right]$	Python $\left[{\mathrm{ms}}_{n}\right]$
2	4.6	4.5	4.5
3	8.3	8.2	8.3
5	16.1	16.0	16.1
10	36.0	36.2	36.2
15	56.6	56.7	56.8
20	76.6	77.3	77.4
30	119.8	118.9	119.0
50	202.4	202.3	202.5
100	411.4	411.6	412.0
150	622.1	621.3	621.9
200	831.2	831.1	832.0
256	1068.8	NA	1067.9

**Table 5 sensors-25-02994-t005:** Simulation settings.

Parameter Name	Value
Simulation area	100 m×100 m
Number of stations	1, 2, 3, 5, 10, 15, 20, 25, 50, 100, 150, 200 and 256
No. of CRIs required to compute $\hat{N}$ in 2CA-R^2^	1
Maximum transmission range ($d_{r\text{\_}max}$)	71 m
Data transmission rate (*R*)	1Mbps
Data packet size	65 bytes
RTS size/REQ size	20 bytes
DCF Retry Limit	7
DTI slot duration	520μs
CRIminislotduration	160μs
Simulationlength (*t_s_*)	60 s

**Table 6 sensors-25-02994-t006:** Statistics related to the initial collision multiplicity using 2CA-R^2^.

Number of Devices	Fraction of CRIs with 1 Device [%]	Fraction of CRIs with 2 Devices [%]	Fraction of CRIs with More than 2 Devices [%]	Channel Utilization [%]
25	99.46	0.53	0.01	4.30
50	98.63	1.34	0.03	8.97
100	96.77	3.10	0.13	18.33
200	91.08	7.48	1.44	42.23
256	87.29	9.62	3.09	57.50

## Data Availability

Data are contained within the article.

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
