# Peer review of "2CA-R2: A Hybrid MAC Protocol for Machine-Type Communications"

_sensors, 2025, doi:10.3390/s25102994_

Round 1
Reviewer 1 Report
Comments and Suggestions for Authors
This paper introduces a novel hybrid Medium Access Control (MAC) protocol named 2CA-R^2, designed to enhance the efficiency of machine-to-machine (M2M) communications in the context of the Internet of Things (IoT). Traditional MAC protocols, such as those based on contention (e.g., CSMA/CA) or reservation (e.g., TDMA), face limitations when handling the unique characteristics of M2M traffic, which often involves a large number of devices generating small, uplink-oriented data packets. The 2CA-R^2 protocol combines the benefits of both contention and reservation strategies by dividing the transmission cycle into two stages: a contention stage using an adaptive version of the 2C conflict-resolution algorithm (Adaptive-2C) and a reservation stage where data is transmitted using time scheduling. The Adaptive-2C algorithm dynamically adjusts the probability of station transmission based on the estimated number of contending stations, significantly improving collision resolution efficiency. The protocol was evaluated through simulations and validated with a Markov chain model, demonstrating superior performance in terms of scaling properties, access delay, throughput, and fairness compared to the widely used DCF protocol in IEEE 802.11 standards.
Some comments are as follows:
1. The authors should keep the consistent usage of terminologies. For example, in the figures, the authors should use "2CA-R^2" instead of "2CA-R2".
2. The introduction could benefit from a more detailed explanation of the specific challenges faced by traditional MAC protocols in M2M communications. For example, elaborating on how the characteristics of M2M traffic (e.g., predominantly uplink data transfer, small packet sizes, and bursty traffic) exacerbate the limitations of existing protocols.
3. The authors should explain why there are different numbers of curves in Figures 10-15. In Figures 10-11, there are results for three methods, but in Figure 12, there is only one method.
4. The method description should be enhanced. The paper provides a detailed explanation of the Adaptive-2C algorithm, but it could be further enhanced by including more visual aids (e.g., flowcharts or diagrams) to illustrate the state transitions and decision-making process. Algorithm pesudocode is also expected.
5. The simulation scenarios could be expanded to include a wider range of conditions, such as varying packet sizes, different data rates, and different network topologies. This would provide a more comprehensive evaluation of the protocol's robustness.
6. While the paper compares 2CA-R2 with DCF, it would be beneficial to include a comparative analysis with other hybrid MAC protocols mentioned in the related work section.
7. The literature review is not comprehensive. Table 1. is only updated to 2022. The authors should give a high-level summary of existing research gaps in the related work discussion.
8. A table for summarizing all mathematical symbols is expected.
9. The conclusion could be expanded to include a summary of the key findings and a discussion of potential future work. For example, the application of artificial intelligence in IoTs by referring to Generative AI for Consumer Internet of Things: Challenges and Opportunities and Agent-based Internet of Things: State-of-the-art and research challenges and how AI can help to improve the performance.
Reviewer 2 Report
Comments and Suggestions for Authors
- Please ensure that all abbreviations in the text (such as Adaptive-2C, 2CA-R2, CRI, DTI, COP/TOP, etc.) are fully defined when they first appear, and use the abbreviations uniformly in the future. For example, in Section 2 "Related Work", COP and TOP can be directly abbreviated when they appear for the second time without repeated explanation.
- It is recommended to add key experimental data (such as throughput improvement percentage, average delay reduction value, etc.) in the abstract to enhance the persuasiveness of the conclusion in a quantitative way
- Section 2 cites multiple hybrid MAC protocol literatures, but does not clearly compare the differences between this paper and them. It is recommended to add the uniqueness of the Adaptive-2C strategy, for example: compared with existing hybrid MAC protocols, how this paper optimizes the performance of large-scale M2M scenarios through dynamic conflict resolution mechanisms.
- The introduction should further explain the core challenges of channel access in M2M communication (such as massive device competition, low latency requirements, etc.), and clarify the necessity of hybrid MAC (CRI/DTI stage division). For example: analyze the limitations of traditional protocols in high-density scenarios and introduce the optimization value of CRI/DTI.
- The introduction emphasizes the large-scale device requirements of M2M, but the article says "considering a network the size of a WLAN or smaller" (line 117), which is contradictory. It is recommended to adjust the statement to ensure that the background is consistent with the experimental design, such as clearly stating that although the protocol is suitable for small and medium-sized networks, the mechanism can be extended to high-density scenarios.
- The article "hybrid MAC protocols are not exempt from technical challenges since they also inherit the limitations of their constituent protocols." (line97-98) points out that hybrid MAC protocols have inherent defects, but the shortcomings of 2CA-R2 are not analyzed later. It is recommended to add discussion to reflect the objectivity of the research.
- The abstract mentions the description of the MAC protocol for throughput, latency and fairness, but the quantitative standard of fairness (such as Jain index, maximum and minimum fairness) is not defined in Section 6.1. It is recommended to add indicators and analyze experimental results to avoid subjective descriptions.
- It is recommended to move Section 4.3 "Example" to before Section 4.1, merge Section 4.1 (CRI) and Section 4.2 (DTI) into a single section, and demonstrate the interaction process through examples to improve the readability of the article.
- It is recommended to add Section 3.3 after Section 3.2 to compare the differences between 2C and Adaptive-2C, list the improvements (such as dynamic threshold adjustment, conflict detection efficiency, etc.), and highlight the innovation.
- Sections 6.3-6.6 compare the protocol of this article with DCF and 2C-R2. Why is 2C-R2 only added in Figures 10 and 11, and 2C-R2 is discarded in subsequent comparisons (such as fig13-15)? Please explain the reasons.
In addition, all the comparison results mentioned the multi-node situation. For example, the original text described Fig. 13 as follows: "As depicted in Fig. 13 (see the dashed red line), approximately 140 stations can transmit their data packet using DCF, while for the same time window, approximately 200 stations are able to transmit using 2CA-R2. That is, at this point, 2CA-R2 achieves an improvement of 42.8% with respect to DCF." Please explain its significance for high-density scenarios of the Internet of Things.
- Make sure that most of the references are up-to-date and closely focus on topics such as hybrid MAC and M2M communication. It is recommended to add the latest research (such as relevant papers in IEEE IoT Journal in the past two years) and check whether the key work is fully cited.
Comments on the Quality of English Language
can be improved
Reviewer 3 Report
Comments and Suggestions for Authors
The manuscript proposed and evaluated 2CA-R2, a hybrid MAC protocol tailored for Machine-to-Machine (M2M) communications, particularly in IoT environments. It combined contention-based and reservation-based approaches to medium access, using a modified Adaptive-2C algorithm during the contention stage and a TDMA-like schedule during the transmission phase. This two-stage hybrid design allowed for contention resolution followed by conflict-free data transmission. The performance of the protocol was evaluated both through OMNeT++ simulations and a Markov chain-based analytical model, showing significant improvements over the IEEE 802.11 DCF in terms of scalability, throughput, access delay, and fairness. The paper is worthwhile contribution, however the authors might find the below comments to improve the manuscript's quality.
1) While the system is tested up to 256 nodes, future IoT systems may involve thousands or millions of nodes. The paper does not clearly address how 2CA-R2 scales in extremely dense deployments. Please elaborate the scalability of the proposed approach.
2) The Markov model assumes error-free channels, and though packet error rates (PER) are briefly discussed, no deep analysis of error resilience (e.g., under interference or fading) is provided. Please justify this assumption or discuss its practicability.
3) The Adaptive-2C algorithm requires estimating the number of contending nodes, which introduces computation and signaling overhead. The complexity and energy costs for resource-constrained IoT devices are not analyzed. Please elaborate this point.
4) Power consumption is a critical metric for M2M/IoT devices, but the paper did not provide any energy consumption or efficiency evaluation, which will be a notable mission. Please the energy consumption issue if possible.
5) Although fairness is claimed as a key strength of 2CA-R2, there was no dedicated fairness metric (e.g., Jain's Index) or comparative evaluation against other schemes beyond qualitative discussion. Please elaborate this point.
6) Since the proposed algorithm is based on TDMA-liked scheduling, please discuss how synchronization can be done among such massive number of IoT sensor nodes.
Round 2
Reviewer 1 Report
Comments and Suggestions for Authors
The authors have resolved most of the comments. However, the suggestion for the conclusion part is not adopted. The current version is too wordy and is lack of specific future direction discussion.
Reviewer 2 Report
Comments and Suggestions for Authors
The authors have improved the manuscript according the reviewer's comments.
Comments on the Quality of English Language
could improved
Author Response
We thank the reviewer for his/her comments and time.
Reviewer 3 Report
Comments and Suggestions for Authors
The authors had responded properly so it can be published at the current stage.
